# EvolveGraph: Multi-Agent Trajectory Prediction with Dynamic Relational Reasoning

**Jiachen Li**[1,2,*,†]   **Fan Yang**[2,*]   **Masayoshi Tomizuka**[2]   **Chiho Choi**[1]

[1]Honda Research Institute, USA    [2] University of California, Berkeley
{jiachen_li, fanyang16, tomizuka}@berkeley.edu   cchoi@honda-ri.com

## Abstract

Multi-agent interacting systems are prevalent in the world, from purely physical systems to complicated social dynamic systems. In many applications, effective understanding of the situation and accurate trajectory prediction of interactive agents play a significant role in downstream tasks, such as decision making and planning. In this paper, we propose a generic trajectory forecasting framework (named EvolveGraph) with explicit relational structure recognition and prediction via latent interaction graphs among multiple heterogeneous, interactive agents. Considering the uncertainty of future behaviors, the model is designed to provide multi-modal prediction hypotheses. Since the underlying interactions may evolve even with abrupt changes, and different modalities of evolution may lead to different outcomes, we address the necessity of dynamic relational reasoning and adaptively evolving the interaction graphs. We also introduce a double-stage training pipeline which not only improves training efficiency and accelerates convergence, but also enhances model performance. The proposed framework is evaluated on both synthetic physics simulations and multiple real-world benchmark datasets in various areas. The experimental results illustrate that our approach achieves state-of-the-art performance in terms of prediction accuracy.

## 1  Introduction

Multi-agent trajectory prediction is critical in many real-world applications, such as autonomous driving, mobile robot navigation and other areas where a group of entities interact with each other, giving rise to complicated behavior patterns at the level of both individuals and the multi-agent system as a whole. Since usually only the trajectories of individual entities are available without any knowledge of the underlying interaction patterns, and there are usually multiple possible modalities for each agent, it is challenging to model such dynamics and forecast their future behaviors.

There have been a number of existing works trying to provide a systematic solution to multi-agent interaction modeling. Some related techniques include, but not limited to social pooling layers [1], attention mechanisms [42, 19, 12, 40, 21], message passing over graphs [8, 37, 22], etc. These techniques can be summarized as implicit interaction modeling by information aggregation. Another line of research is to explicitly perform inference over the structure of the latent interaction graph, which allows for relational structures with multiple interaction types [18, 2]. Our approach falls into this category but with significant extension and performance enhancement over existing methods.

A closely related work is NRI [18], in which the interaction graph is static with homogeneous nodes during training. This is sufficient for the systems involving homogeneous type of agents with fixed

interaction patterns. In many real-world scenarios, however, the underlying interactions are inherently varying even with abrupt changes (e.g. basketball players). And there may be heterogeneous types of agents (e.g. cars, pedestrians, cyclists, etc.) involved in the system, while NRI cannot distinguish them explicitly. Moreover, NRI does not deal with the multi-modality explicitly in future system behaviors. In this work, we address the problem of 1) extracting the underlying interaction patterns with a latent graph structure, which is able to handle different types of agents in a unified way, 2) capturing the dynamics of interaction graph evolution for dynamic relational reasoning, 3) predicting future trajectories (state sequences) based on the historical observations and the latent interaction graph, and 4) capturing the multi-modality of future system behaviors.

The main contributions of this paper are summarized as:

- We propose a generic trajectory forecasting framework with explicit interaction modeling via a latent graph among multiple heterogeneous, interactive agents. Both trajectory information and context information (e.g. scene images, semantic maps, point cloud density maps) can be incorporated into the system.

- We propose a dynamic mechanism to evolve the underlying interaction graph adaptively along time, which captures the dynamics of interaction patterns among multiple agents. We also introduce a double-stage training pipeline which not only improves training efficiency and accelerates convergence, but also enhances model performance in terms of prediction accuracy.

- The proposed framework is designed to capture the uncertainty and multi-modality of future trajectories in nature from multiple aspects.

- We validate the proposed framework on both synthetic simulations and trajectory forecasting benchmarks in different areas. Our EvolveGraph achieves the state-of-the-art performance consistently.

## 2  Related work

The problem of multi-agent trajectory prediction has been considered as modeling behaviors among a group of interactive agents. Social forces was introduced by [10] to model the attractive and repulsive motion of humans with respect to the neighbors. Some other learning-based approaches were proposed, such as hidden Markov models [23, 46], dynamic Bayesian networks [16], inverse reinforcement learning [39]. In recent years, the conceptual extension has been made to better model social behavior with supplemental cues such as motion patterns [48, 45] and group attributes [44]. Such social models have motivated the recent data-driven methods in [1, 20, 7, 43, 9, 47, 11, 24, 4, 50, 26, 34, 38, 32, 21, 6, 13, 28, 25]. They encode the motion history of individual entities using the recurrent operation of neural networks. However, it is nontrivial for these methods to find acceptable future motions in heterogeneous and interactively changing environments, partly due to their heuristic feature pooling or aggregation, which may not be sufficient for dynamic interaction modeling.

Interaction modeling and relational reasoning have been widely studied in various fields. Recently, deep neural networks applied to graph structures have been employed to formulate a connection between interactive agents or variables [42, 26, 19, 22, 36, 49, 3]. These methods introduce nodes to represent interactive agents and edges to express their interactions with each other. They directly learn the evolving dynamics of node attributes (agents' states) and/or edge attributes (relations between agents) by constructing spatio-temporal graphs. However, their models have no explicit knowledge about the underlying interaction patterns. Some existing works (e.g. NRI [18]) have taken a step forward towards explicit relational reasoning by inferring a latent interaction graph. However, it is nontrivial for NRI to deal with heterogeneous agents, context information and the systems with varying interactions. In this work, we present an effective solution to handle aforementioned issues. Our work is also related to learning on dynamic graphs. Most existing works studied representation learning on dynamically evolving graphs [30, 17], while we attempt to predict evolution of the graph.

## 3  Problem formulation

We assume that, without loss of generality, there are $N$ homogeneous or heterogeneous agents in the scene, which belongs to $M$ ($\geq 1$) categories (e.g. cars, cyclists, pedestrians). The number of

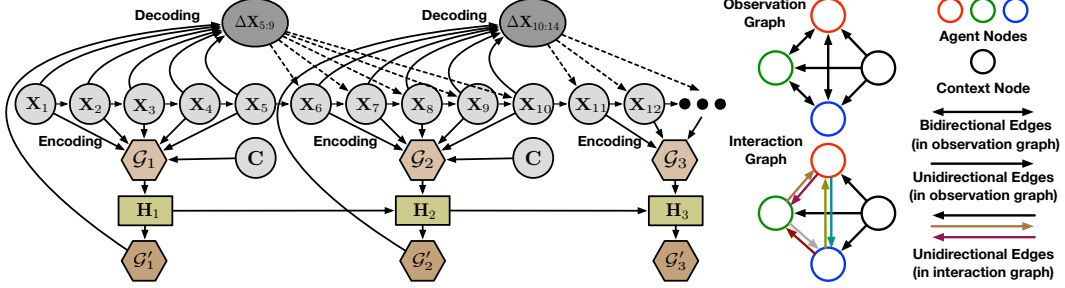

Figure 1: (a) The left part is a high-level graphical illustration of the proposed approach, where the encoding horizon and decoding horizons (re-encoding gap) are both set to 5. $\mathbf{X}_t$ denotes the state of all the agents at time $t$, $\Delta\mathbf{X}_t$ denotes the change in state, and $\mathbf{C}$ denotes context information. $\mathcal{G}_\beta$ denotes the latent interaction graph obtained from the static encoding process, and $\mathcal{G}'_\beta$ denotes the adjusted interaction graph with time dependence. At each encoding-decoding iteration, $\mathcal{G}_\beta$ is obtained through the encoding of previous trajectories and context information, which goes through a recurrent unit ($\mathbf{H}_\beta$) to get the adjusted interaction graph $\mathcal{G}'_\beta$. The previous trajectories and $\mathcal{G}'_\beta$ are combined as the input of the decoding process, which generates distributions of state changes to get future trajectories. (b) The right part is an illustration of observation graph and interaction graph. In the observation graph, the edges between agent nodes are homogeneous and bidirectional, while in the interaction graph, colored edges are unidirectional with a certain type. (Best viewed in color.)

agents may vary in different cases. We denote a set of state sequences covering the historical and forecasting horizons ($T_h$ and $T_f$) as $\mathbf{X}_{1:T} = \{\mathbf{x}^i_{1:T}, T = T_h + T_f, i = 1, ..., N\}$. We also denote a sequence of historical context information as $\mathbf{C}_{1:T_h} = \{\mathbf{c}_{1:T_h}\}$ for dynamic scenes or fixed context information $\mathbf{C}$ for static scenes. In the scope of this paper, we define $\mathbf{x}^i_t = (x^i_t, y^i_t)$, where $(x, y)$ is the 2D coordinate in the world space or image pixel space. The context information includes images or tensors which represent attributes of the scene. We denote the latent interaction graph as $\mathcal{G}_\beta$, where $\beta$ is the graph index. We aim to estimate $p(\mathbf{X}_{T_h+1:T_h+T_f}|\mathbf{X}_{1:T_h}, \mathbf{C}_{1:T_h})$ for dynamic scenes or $p(\mathbf{X}_{T_h+1:T_h+T_f}|\mathbf{X}_{1:T_h}, \mathbf{C})$ for static scenes. For simplicity, we use $\mathbf{C}$ when referring to the context information in the equations. More formally, if the latent interaction graph is inferred at each time step, then we have the factorization of $p(\mathbf{X}_{T_h+1:T_h+T_f}|\mathbf{X}_{1:T_h}, \mathbf{C})$ below:

$$\int_{\mathcal{G}} p(\mathcal{G}_0|\mathbf{X}_{1:T_h}, \mathbf{C})p(\mathbf{X}_{T_h+1}|\mathcal{G}_0, \mathbf{X}_{1:T_h}, \mathbf{C}) \prod_{\beta=1}^{T_f-1} p(\mathcal{G}_\beta|\mathcal{G}_{0:\beta-1}, \mathbf{X}_{1:T_h+\beta}, \mathbf{C})p(\mathbf{X}_{T_h+\beta+1}|\mathcal{G}_{0:\beta}, \mathbf{X}_{1:T_h+\beta}, \mathbf{C}).$$

## 4 EvolveGraph

An illustrative graphical model is shown in Figure 1 (left part) to demonstrate the essential procedures of the prediction framework with explicit dynamic relational reasoning. Instead of end-to-end training in a single pipeline, our training process contains two consecutive stages:

• *Static interaction graph learning*: A series of encoding functions are trained to extract interaction patterns from the observed trajectories and context information, and generate a distribution of static latent interaction graphs. A series of decoding functions are trained to recurrently generate multi-modal distributions of future states. At this stage, the prediction is only based on the static interaction graph inferred from the history information, which means the encoding process is only applied once and the interaction graph does not evolve with the decoding process.

• *Dynamic interaction graph learning*: The pre-trained encoding and decoding functions at the first stage are utilized as an initialization, which are finetuned together with the training of a recurrent network which captures the dynamics of interaction graph evolution. The graph recurrent network serves as a high-level integration which considers the dependency of the current interaction graph on previous ones. At this stage, the prediction is based on the latest updated interaction graph.

### 4.1 Static interaction graph learning

**Observation Graph** A fully-connected graph without self-loops is constructed to represent the observed information with node/edge attributes, which is called observation graph. Assume that

there are $N$ heterogeneous agents in the scene, which belongs to $M$ categories. Then the observation graph consists of $N$ agent nodes and one context node. Agent nodes are bidirectionally connected to each other, and the context node only have outgoing edges to each agent node. We denote an observation graph as $\mathcal{G}_{obs} = \{\mathcal{V}_{obs}, \mathcal{E}_{obs}\}$, where $\mathcal{V}_{obs} = \{\mathbf{v}_i, i \in \{1, ..., N\}\} \cup \{\mathbf{v}_c\}$ and $\mathcal{E}_{obs} = \{\mathbf{e}_{ij}, i, j \in \{1, ..., N\}\} \cup \{\mathbf{e}_{ic}, i \in \{1, ..., N\}\}$ . $\mathbf{v}_i$, $\mathbf{v}_c$ and $\mathbf{e}_{ij}$, $\mathbf{e}_{ic}$ denote agent node attribute, context node attribute and agent-agent, context-agent edge attribute, respectively. More specifically, the $\mathbf{e}_{ij}$ denotes the attribute of the edge from node $j$ to node $i$. Each agent node has two types of attributes: *self-attribute* and *social-attribute*. The former only contains the node's own state information, while the latter only contains other nodes' state information. The calculations of node/edge attributes are given by

$$\mathbf{v}_i^{\text{self}} = f_a^m(\mathbf{x}_{1:T_h}^i), \ i \in \{1, ..., N\}, \ m \in \{1, ..., M\}, \quad \mathbf{v}_c = f_c(\mathbf{c}_{1:T_h}) \quad \text{or} \quad \mathbf{v}_c = f_c(\mathbf{c}), \tag{1}$$

$$\mathbf{e}_{ij}^1 = f_e^1([\mathbf{v}_i^{\text{self}}, \mathbf{v}_j^{\text{self}}]), \ \mathbf{e}_{ic}^1 = f_{ec}^1([\mathbf{v}_i^{\text{self}}, \mathbf{v}_c]), \ \mathbf{v}_i^{\text{social-1}} = f_v^1([\sum_{i \neq j} \alpha_{ij}\mathbf{e}_{ij}^1, \ \mathbf{e}_{ic}^1]), \ \sum_{i \neq j} \alpha_{ij} = 1, \tag{2}$$

$$\mathbf{v}_i^1 = [\mathbf{v}_i^{\text{self}}, \mathbf{v}_i^{\text{social-1}}], \quad \mathbf{e}_{ij}^2 = f_e^2([\mathbf{v}_i^1, \mathbf{v}_j^1]), \quad \alpha_{ij} = \frac{\exp\left(\text{LeakyReLU}(\mathbf{a}^\top[\mathbf{W}\mathbf{v}_i||\mathbf{W}\mathbf{v}_j])\right)}{\sum_{k \in \mathcal{N}_i} \exp\left(\text{LeakyReLU}(\mathbf{a}^\top[\mathbf{W}\mathbf{v}_i||\mathbf{W}\mathbf{v}_k])\right)}, \tag{3}$$

where $\alpha_{ij}$ are learnable attention coefficients computed similar as [41], $f_a^m(\cdot)$, $f_c(\cdot)$ are agent, context node embedding functions, and $f_e(\cdot)$, $f_{ec}(\cdot)$ and $f_v(\cdot)$ are agent-agent edge, agent-context edge, and agent node update functions, respectively. Different types of nodes (agents) use different embedding functions. Note that the attributes of the context node are never updated and the edge attributes only serve as intermediates for the update of agent node attributes. These $f(\cdot)$ functions are implemented by deep networks with proper architectures. At this time, we obtain a complete set of node/edge attributes which include the information of direct (first-order) interaction. The higher-order interactions can be modeled by multiple loops of equations (2)-(3), in which the social node attributes and edge attributes are updated by turns. Note that the *self-attribute* is fixed in the whole process.

**Interaction Graph** The interaction graph represents interaction patterns with a distribution of edge types for each edge, which is built on top of the observation graph. We set a hyperparameter $L$ to denote the number of possible edge types (interaction types) between pairwise agent nodes to model *agent-agent* interactions. Also, there is another edge type that is shared between the context node and all agent nodes to model *agent-context* interactions. Note that "no edge" can also be treated as a special edge type, which implies that there is no message passing along such edges. More formally, the interaction graph is a discrete probability distribution $q(\mathcal{G}|\mathbf{X}_{1:T_h}, \mathbf{C}_{1:T_h})$ or $q(\mathcal{G}|\mathbf{X}_{1:T_h}, \mathbf{C})$, where $\mathcal{G} = \{\mathbf{z}_{ij}, i, j \in \{1, ..., N\}\} \cup \{\mathbf{z}_{ic}, i \in \{1, ..., N\}\}$ is a set of interaction types for all the edges, and $\mathbf{z}_{ij}$ and $\mathbf{z}_{ic}$ are random variables to indicate pairwise interaction types for a specific edge.

**Encoding** The goal of the encoding process is to infer a latent interaction graph from the observation graph, which is essentially a multi-class edge classification task. We employ a softmax function with a continuous approximation of the discrete distribution [27] on the last updated edge attributes to obtain the probability of each edge type, which is given by

$$q(\mathbf{z}_{ij}|\mathbf{X}_{1:T_h}, \mathbf{C}) = \text{Softmax}((\mathbf{e}_{ij}^2 + \mathbf{g})/\tau), \ i, j \in \{1, ..., N\}, \tag{4}$$

where $\mathbf{g}$ is a vector of independent and identically distributed samples drawn from Gumbel$(0, 1)$ distribution and $\tau$ is the Softmax temperature, which controls the sample smoothness. We also use the repramatrization trick to obtain gradients for backpropagation. The edge type between context node and agent nodes $\mathbf{z}_{ic}$, without loss of generality, is hard-coded with probability one. For simplicity, we summarize all the operations in the observation graph and the encoding process as $q(\mathbf{z}|\mathbf{X}_{1:T_h}, \mathbf{C}) = f_{\text{enc}}(\mathbf{X}_{1:T_h}, \mathbf{C})$, which gives a factorized distribution of $\mathbf{z}_{ij}$.

**Decoding** Since in many real-world applications the state of agents has long-term dependence, a recurrent decoding process is applied to the interaction graph and observation graph to approximate the distribution of future trajectories $p(\mathbf{X}_{T_h+1:T_h+T_f}|\mathcal{G}, \mathbf{X}_{1:T_h}, \mathbf{C})$. The output at each time step is a Gaussian mixture distribution with $K$ components, where the covariance of each Gaussian component is manually set equal. The detailed operations in the decoding process consists of two stages: burn-in stage ($1 \leq t \leq T_h$) and prediction stage ($T_h + 1 \leq t \leq T_h + T_f$), which are given by

$$\tilde{\mathbf{e}}_t^{ij} = \sum_{l=1}^{L} z_{ij,l} \tilde{f}_e^l([\tilde{\mathbf{h}}_t^i, \tilde{\mathbf{h}}_t^j]), \quad \text{MSG}_t^i = \sum_{j \neq i} \tilde{\mathbf{e}}_t^{ij}, \tag{5}$$

- $1 \leq t \leq T_h$ (Burn-in stage):

$$\tilde{\mathbf{h}}_{t+1}^i = \text{GRU}^i([\text{MSG}_t^i, \mathbf{x}_t^i, \mathbf{v}_c], \tilde{\mathbf{h}}_t^i), \quad w_{t+1}^{i,k} = f_{weight}^k(\tilde{\mathbf{h}}_{t+1}^i), \tag{6}$$

$$\boldsymbol{\mu}_{t+1}^{i,k} = \mathbf{x}_t^i + f_{\text{out}}^k(\tilde{\mathbf{h}}_{t+1}^i), \quad p(\hat{\mathbf{x}}_{t+1}^i|\mathbf{z}, \mathbf{x}_{1:t}^i, \mathbf{c}) = \sum_{k=1}^{K} w_{t+1}^{i,k} \mathcal{N}(\boldsymbol{\mu}_{t+1}^{i,k}, \sigma^2 \mathbf{I}), \tag{7}$$

- $T_h + 1 \leq t \leq T_h + T_f$ (Prediction stage):

$$\tilde{\mathbf{h}}_{t+1}^i = \text{GRU}^i([\text{MSG}_t^i, \hat{\mathbf{x}}_t^i, \mathbf{v}_c], \tilde{\mathbf{h}}_t^i), \quad w_{t+1}^{i,k} = f_{weight}^k(\tilde{\mathbf{h}}_{t+1}^i), \quad \boldsymbol{\mu}_{t+1}^{i,k} = \hat{\mathbf{x}}_t^i + f_{\text{out}}^k(\tilde{\mathbf{h}}_{t+1}^i), \qquad (8)$$

$$p(\hat{\mathbf{x}}_{t+1}^i | \mathbf{z}, \hat{\mathbf{x}}_{T_h+1:t}^i, \mathbf{x}_{1:T_h}^i, \mathbf{c}) = \sum_{k=1}^{K} w_{t+1}^{i,k} \mathcal{N}(\boldsymbol{\mu}_{t+1}^{i,k}, \sigma^2 \mathbf{I}), \qquad (9)$$

Sample a Gaussian component from the mixture based on $\mathbf{w}_{t+1}^i$, $\quad$ Set $\hat{\mathbf{x}}_{t+1}^i = \boldsymbol{\mu}_{t+1}^{i,k}$, $\qquad (10)$

where MSG is a symbolic acronym for "message" here without specific meanings, $\tilde{\mathbf{h}}_t^i$ is the hidden state of $\text{GRU}^i$ at time $t$, $w_{t+1}^{i,k}$ is the weight of the $k$th Gaussian distribution at time step $t+1$ for agent $i$. $\tilde{f}_e^l(\cdot)$ is the edge update function of edge type $l$, $f_{weight}^k(\cdot)$ is a mapping function to get the weight of the $k$th Gaussian distribution, and $f_{\text{out}}^k(\cdot)$ is a mapping function to get the mean of the $k$th Gaussian component. Note that the predicted $\hat{\mathbf{x}}_t^i$ is needed in equation (8). In the previous decoding step, we only have its corresponding distribution $p(\hat{\mathbf{x}}_t^i | \mathbf{z}, \hat{\mathbf{x}}_{T_h+1:t-1}^i, \mathbf{x}_{1:T_h}^i, \mathbf{c})$ from the previous step. We first sample a Gaussian component from the mixture based on the component weights $w_{t+1}^i$. Say we get the $k$th component, then we set $\hat{\mathbf{x}}_t^i$ as $\boldsymbol{\mu}_t^{j,k}$, which is the trajectory with maximum likelihood within this component. We fix the covariance $\sigma$ as a constant. The nodes (agents) of the same type share the same GRU decoder. During the burn-in stage, the ground-truth states are used; while during the prediction stage, the state prediction hypotheses are used as the input at the next time step iteratively. For simplicity, the whole decoding process is summarized as $p(\mathbf{X}_{T_h+1:T_h+T_f} | \mathcal{G}, \mathbf{X}_{1:T_h}, \mathbf{C}) = f_{\text{dec}}(\mathcal{G}, \mathbf{X}_{1:T_h}, \mathbf{C})$.

## 4.2 Dynamic interaction graph

In many applications, the interaction patterns recognized from the past time steps are likely not static in the future. Instead, they are rather dynamically evolving throughout the future time steps. Moreover, many interaction systems have multi-modal properties in its nature. Different modalities afterwards are likely to result in different interaction patterns. A single static interaction graph is neither sufficiently flexible to model dynamically changing situations (especially those with abrupt changes), nor to capture all the modalities. Therefore, we introduce an effective dynamic mechanism to evolve the interaction graph.

The encoding process is repeated every $\tau$ (re-encoding gap) time steps to obtain the latent interaction graph based on the latest observation graph. Since the new interaction graph also has dependence on previous ones, we also need to consider their effects. Therefore, a recurrent unit (GRU) is utilized to maintain and propagate the history information, as well as adjust the prior interaction graphs. More formally, the calculations are given by

$$q(\mathbf{z}_\beta | \mathbf{X}_{1+\beta\tau:T_h+\beta\tau}, \mathbf{C}) = f_{\text{enc}}(\mathbf{X}_{1+\beta\tau:T_h+\beta\tau}, \mathbf{C}), \qquad (11)$$

$$q(\mathbf{z}_\beta' | \mathbf{X}_{1+\beta\tau:T_h+\beta\tau}, \mathbf{C}) = \text{GRU}(q(\mathbf{z}_\beta | \mathbf{X}_{1+\beta\tau:T_h+\beta\tau}, \mathbf{C}), \mathbf{H}_\beta) \qquad (12)$$

where $\beta$ is the re-encoding index starting from 0, $\mathbf{z}_\beta$ is the interaction graph obtained from the static encoding process, $\mathbf{z}_\beta'$ is the adjusted interaction graph with time dependence, and $\mathbf{H}_\beta$ is the hidden state of the graph evolution GRU. After obtaining $\mathcal{G}_\beta' = \{\mathbf{z}_\beta'\}$, the decoding process is applied to get the states of the next $\tau$ time steps,

$$p(\mathbf{X}_{T_h+\beta\tau+1:T_h+(\beta+1)\tau} | \mathcal{G}_\beta', \mathbf{X}_{1:T_h}, \hat{\mathbf{X}}_{T_h+1:T_h+\beta\tau}, \mathbf{C}) = f_{\text{dec}}(\mathcal{G}_\beta', \mathbf{X}_{1:T_h}, \hat{\mathbf{X}}_{T_h+1:T_h+\beta\tau}, \mathbf{C}). \qquad (13)$$

The decoding and re-encoding processes are iterated to obtain the distribution of future trajectories.

## 4.3 Uncertainty and multi-modality

Here we emphasize the efforts to encourage diverse and multi-modal trajectory prediction and generation. In our framework, the uncertainty and multi-modality mainly come from three aspects. First, in the decoding process, we output Gaussian mixture distributions indicating that there are several possible modalities at the next step. We only sample a single Gaussian component at each step based on the component weights which indicate the probability of each modality. Second, different sampled trajectories will lead to different interaction graph evolution. Evolution of interaction graphs contributes to the multi-modality of future behaviors, since different underlying relational structures enforce different regulations on the system behavior and lead to various outcomes. Third, directly training such a model, however, tends to collapse to a single mode. Therefore, we employ an effective mechanism to mitigate the mode collapse issue and encourage multi-modality. During training,

Table 1: Comparison of Accuracy (Mean $\pm$ Std in %) of Interaction (Edge Type) Recognition.

| | Corr. (LSTM) | NRI (dynamic) | **EvolveGraph** (static) | **EvolveGraph** (RNN re-encoding) | **EvolveGraph** (dynamic) | Supervised |
|---|---|---|---|---|---|---|
| No Change | 63.2$\pm$0.9 | 91.3$\pm$0.3 | **95.6**$\pm$0.2 | 91.4$\pm$0.3 | 93.8$\pm$1.1 | 98.1$\pm$0.4 |
| Change | — | 71.5$\pm$3.1 | 64.1$\pm$0.8 | 75.2$\pm$1.4 | **82.3**$\pm$3.2 | 94.3$\pm$1.5 |

we run the decoding process $d$ times, which generates $d$ trajectories for each agent under specific scenarios. We only choose the prediction hypothesis with the minimal loss for backpropagation, which is the most likely to be in the same mode as the ground truth. The other prediction hypotheses may have much higher loss, but it doesn't necessarily imply that they are implausible. They may represent other potential reasonable modalities.

### 4.4 Loss Function and Training

In our experiments, we first train the encoding / decoding functions using a static interaction graph. Then in the process of training dynamic interaction graph, we use the pre-trained encoding / decoding functions at the first stage to initialize the parameters of the modules used in the dynamic training. This step is reasonable since the encoding / decoding functions used in these two training process play similar roles and their optima are supposed to be close. And if we train dynamic graphs directly, it will lead to longer convergence time and is likely to be trapped into some bad local optima due to large number of learnable parameters. It is possible that this method may accelerate the whole training process and avoid some bad local optima.

In the training process, our loss function is defined as follows:

$$\mathcal{L}_S = -\mathbb{E}_{q(\mathbf{z}|\mathbf{X}_{1:T_h},\mathbf{C})}\left[\sum_{i=1}^{N}\sum_{t=T_h+1}^{T_h+T_f}\sum_{k=1}^{K} w_t^{i,k} \log p_t^{i,k}(\mathbf{x}_t|\mathbf{z},\mathbf{X}_{1:T_h},\hat{\mathbf{X}}_{T_h+1:t-1},\mathbf{C})\right] \text{ (Static)}, \tag{14}$$

$$\mathcal{L}_D = -\mathbb{E}_{q(\mathbf{z}'_{\beta(t)}|\mathbf{X}_{1+\beta\tau:T_h+\beta\tau},\mathbf{C})}\left[\sum_{i=1}^{N}\sum_{t=T_h+1}^{T_h+T_f}\sum_{k=1}^{K} w_t^{i,k} \log p_t^{i,k}(\mathbf{x}_t|\mathbf{z}'_{\beta(t)},\mathbf{X}_{1:T_h},\hat{\mathbf{X}}_{T_h+1:t-1},\mathbf{C})\right] \text{ (Dynamic)}, \tag{15}$$

where $q(\cdot)$ denotes the encoding and re-encoding operations, which return a factorized distribution of $\mathbf{z}_{ij}$ or $\mathbf{z}'_{ij}$. The $p_t^{i,k}(\mathbf{x}_t|\mathbf{z},\mathbf{X}_{1:T_h},\hat{\mathbf{X}}_{T_h+1:t-1},\mathbf{C})$ and $p_t^{i,k}(\mathbf{x}_t|\mathbf{z}'_{\beta(t)},\mathbf{X}_{1:T_h},\hat{\mathbf{X}}_{T_h+1:t-1},\mathbf{C})$ denote a certain Gaussian distribution.

## 5 Experiments

In this paper, we validated the proposed framework EvolveGraph on one synthetic dataset and three benchmark datasets for real-world applications: Honda 3D Dataset (H3D) [31], NBA SportVU Dataset (NBA), and Stanford Drone Dataset (SDD) [33]. The dataset details, baseline approaches, as well as implementation details are introduced in supplementary materials.

For the synthetic dataset, since we have access to the ground truth of the underlying interaction graph, we quantitatively and qualitatively evaluate the model performance in terms of both interaction (edge type) recognition and average state prediction error. For the benchmark datasets, we evaluate the model performance in terms of two widely used standard metrics: minimum average displacement error (minADE$_{20}$) and minimum final displacement error (minFDE$_{20}$) [4]. The minADE$_{20}$ is defined as the minimum average distance between the 20 predicted trajectories and the ground truth over all the involved entities within the prediction horizon. The minFDE$_{20}$ is defined as the minimum deviated distance of 20 predicted trajectories at the last predicted time step. We also provide ablative analysis (right part of Table 2-4), analysis on double-stage training, analysis on the selection of edge types and re-encoding gap, and additional qualitative results in supplementary materials.

### 5.1 Synthetic simulations: particle physics system

We experimented with a simulated particle system with change of relations. Multiple particles are initially linked and move together. The links disappear as long as a certain criterion on particle state is satisfied and the particles move independently thereafter. The model is expected to learn the

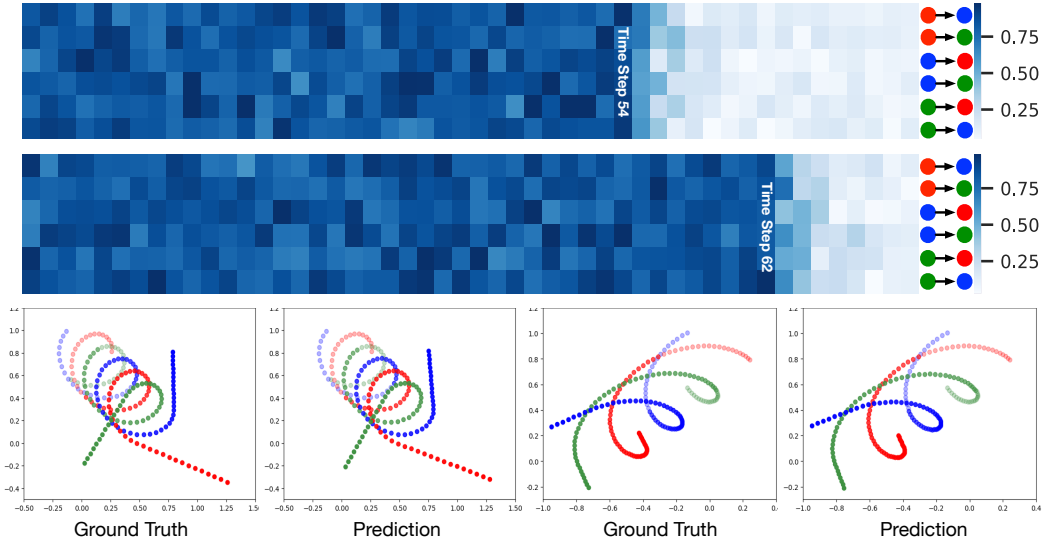

Figure 2: Visualization of latent interaction graph evolution and particle trajectories. (a) The top two figures show the probability of the first edge type ("with link") at each time step. Each row corresponds to a certain edge (shown in the right). The actual times of graph evolution are 54 and 62, respectively. The model is able to capture the underlying criterion of relation change and further predict the change of edge types with nearly no delay. (b) The figures in the last row show trajectory prediction results, where semi-transparent dots are historical observations.

criterion by itself, and perform edge type prediction and trajectory prediction. Since the system is deterministic in nature, we do not consider multi-modality in this task. Further details on the dataset generation are introduced in Section 8.1.1 in the supplementary materials.

We predicted the particle states at the future 50 time steps based on the observations of 20 time steps. We set two edge types in this task, which correspond to "with link" and "without link". The results of edge type prediction are summarized in Table 1, which are averaged over 3 independent runs. *No Change* means the underlying interaction structure keeps the same in the whole horizon, while *Change* means the change of interaction patterns happens at some time. It shows that the supervised learning baseline, which directly trains the encoding functions with ground truth labels, performs the best in both setups and serves as a "gold standard". Under the *No Change* setup, NRI (dynamic) is compara-

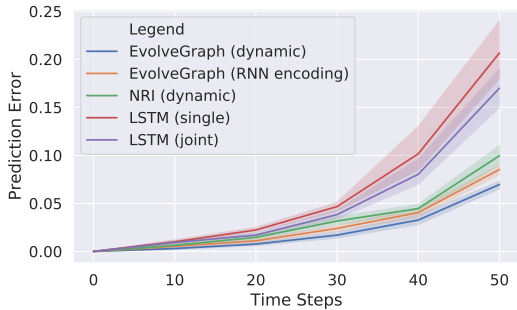

Figure 3: Average error of particle state.

ble to EvolveGraph (RNN re-encoding), while EvolveGraph (static) achieves the best performance. The reason is that dynamic evolution of interaction graph leads to higher flexibility but may result in larger uncertainty, which affects edge prediction in the systems with static relational structures. Under the *Change* setup, NRI (dynamic) re-evaluates the latent graph at every time step during the testing phase, but it is hard to capture the dependency between consecutive graphs, and the encoding functions may not be flexible enough to capture the evolution. EvolveGraph (RNN re-encoding) performs better since it considers the dependency of consecutive steps during the training phase, but it still captures the evolution only at the feature level instead of the graph level. EvolveGraph (dynamic) achieves significantly higher accuracy than the other baselines (except Supervised), due to the explicit evolution of interaction graphs.

We also provide visualization of interaction graphs and particle trajectories of random testing cases in Figure 2. In the heatmaps, despite that the predicted probabilities fluctuate within a small range at each step, they are very close to the ground truth (1 for "with link" and 0 for "without link").

Table 2: minADE$_{20}$ / minFDE$_{20}$ (Meters) of Trajectory Prediction (H3D dataset).

| Time | Baseline Methods | | | | | | | EvolveGraph (Ours) | | | | |
| --- | --- | --- | --- | --- | --- | --- | --- | --- | --- | --- | --- | --- |
| | STGAT | Social-Attention | Social-STGCNN | Social-GAN | Gated-RN | Trajectron++ | NRI (dynamic) | SG (same node type) | SG | RNN re-encoding | DG (single stage) | DG (double stage) |
| 1.0s | 0.24 / 0.33 | 0.29 / 0.45 | 0.23 / 0.32 | 0.27 / 0.37 | **0.18** / 0.32 | 0.21 / 0.34 | 0.24 / 0.30 | 0.28 / 0.37 | 0.27 / 0.35 | 0.25 / 0.32 | 0.24 / 0.31 | 0.19 / **0.25** |
| 2.0s | 0.34 / 0.48 | 0.53 / 0.96 | 0.36 / 0.52 | 0.45 / 0.77 | 0.32 / 0.64 | 0.33 / 0.62 | 0.32 / 0.60 | 0.40 / 0.58 | 0.38 / 0.55 | 0.35 / 0.51 | 0.33 / 0.46 | **0.31** / **0.44** |
| 3.0s | 0.46 / 0.77 | 0.87 / 1.62 | 0.49 / 0.89 | 0.68 / 1.29 | 0.49 / 1.03 | 0.46 / 0.93 | 0.48 / 0.94 | 0.51 / 0.80 | 0.48 / 0.76 | 0.44 / 0.70 | 0.40 / 0.60 | **0.39** / **0.58** |
| 4.0s | 0.60 / 1.18 | 1.21 / 2.56 | 0.73 / 1.49 | 0.94 / 1.91 | 0.69 / 1.56 | 0.71 / 1.63 | 0.73 / 1.56 | 0.64 / 1.21 | 0.61 / 1.14 | 0.57 / 1.07 | 0.50 / 0.90 | **0.48** / **0.86** |

Table 3: minADE$_{20}$ / minFDE$_{20}$ (Meters) of Trajectory Prediction (NBA dataset).

| Time | Baseline Methods | | | | | | | EvolveGraph (Ours) | | | | |
| --- | --- | --- | --- | --- | --- | --- | --- | --- | --- | --- | --- | --- |
| | STGAT | Social-STGCNN | Social-Attention | Social-LSTM | Social-GAN | Trajectron++ | NRI (dynamic) | SG (same node type) | SG | RNN re-encoding | DG (single stage) | DG (double stage) |
| 1.0s | 0.42 / 0.71 | 0.46 / 0.76 | 0.87 / 1.36 | 0.92 / 1.34 | 0.82 / 1.25 | 0.55 / 0.90 | 0.60 / 0.87 | 0.70 / 1.09 | 0.59 / 0.92 | 0.58 / 0.89 | 0.48 / 0.76 | **0.31** / **0.52** |
| 2.0s | 0.91 / 1.39 | 0.90 / 1.43 | 1.58 / 2.51 | 1.64 / 2.74 | 1.52 / 2.45 | 0.99 / 1.58 | 1.02 / 1.71 | 1.51 / 2.38 | 1.38 / 2.12 | 1.09 / 1.88 | 0.84 / 1.43 | **0.74** / **1.10** |
| 3.0s | 1.62 / 2.87 | 1.59 / 2.67 | 2.78 / 4.66 | 2.93 / 5.03 | 2.63 / 4.51 | 1.89 / 3.32 | 1.83 / 3.15 | 2.10 / 3.53 | 1.88 / 3.23 | 1.77 / 2.87 | 1.43 / 2.55 | **1.28** / **2.07** |
| 4.0s | 2.47 / 3.86 | 2.35 / 3.71 | 3.76 / 6.64 | 4.00 / 7.12 | 3.60 / 6.24 | 2.62 / 4.70 | 2.48 / 4.30 | 2.83 / 4.85 | 2.52 / 4.57 | 2.39 / 3.89 | 2.08 / 3.74 | **1.83** / **3.16** |

Table 4: minADE$_{20}$ / minFDE$_{20}$ (Pixels) of Trajectory Prediction (SDD dataset).

| Time | Baseline Methods | | | | | | | EvolveGraph (Ours) | | | | |
| --- | --- | --- | --- | --- | --- | --- | --- | --- | --- | --- | --- | --- |
| | STGAT | Social-STGCNN | Social-Attention | Social-LSTM | Social-GAN | Trajectron++ | NRI (dynamic) | SG (same node type) | SG | RNN re-encoding | DG (single stage) | DG (double stage) |
| 4.8s | 18.8 / 31.3 | 20.6 / 33.1 | 33.3 / 55.9 | 31.4 / 55.6 | 27.0 / 43.9 | 19.3 / 32.7 | 25.6 / 43.7 | 22.5 / 40.3 | 20.6 / 36.4 | 18.4 / 32.1 | 16.1 / 26.6 | **13.9** / **22.9** |

The change of relation can be quickly captured within two time steps. The results of particle state prediction are shown in Figure 3. The standard deviation was calculated over 3 runs. Within the whole horizon, EvolveGraph (dynamic) consistently outperforms the other baselines with stable performance (small standard deviation).

## 5.2 H3D dataset: traffic scenarios

We predicted the future 10 time steps (4.0s) based on the historical 5 time steps (2.0s). The comparison of quantitative results is shown in Table 2, where the unit of reported minADE$_{20}$ and minFDE$_{20}$ is meters in the world coordinates. Note that we included cars, trucks, cyclists and pedestrians in the experiments. All the baseline methods consider the relations and interactions among agents. The Social-Attention employs spatial attention mechanisms, while the Social-GAN demonstrates a deep generative model which learns the data distribution to generate human-like trajectories. The Gated-RN and Trajectron++ both leverage spatio-temporal information to involve relational reasoning, which leads to smaller prediction error. The NRI infers a latent interaction graph and learns the dynamics of agents, which achieves similar performance to Trajectron++. The STGAT and Social-STGCNN further take advantage of the graph neural network to extract relational features in the multi-agent setting. Our proposed method achieves the best performance, which implies the advantages of explicit interaction modeling via evolving interaction graphs. The 4.0s minADE$_{20}$ / minFDE$_{20}$ are *significantly* reduced by 20.0% / 27.1% compared to the best baseline approach (STGAT).

We also provide visualization of results. Figure 4(a) and Figure 4(b) show two random testing samples from H3D results. We can tell that our framework can generate accurate and plausible trajectories. More specifically, in Figure 4(a), for the blue prediction hypothesis at the left bottom, there is an abrupt change at the fifth prediction step. This is because the interaction graph evolved at this step (Our re-encoding gap $\tau$ was set to be 5 in this case). Moreover, in the heatmap, there are multiple possible trajectories starting from this point, which represent multiple potential modalities. These results show that the evolving interaction graph can reinforce the multi-modal property of our model, since different samples of trajectories at the previous steps lead to different directions of graph evolution, which significantly influences the prediction afterwards. In Figure 4(b), each car may leave the roundabout at any exit. Our model can successfully show the modalities of exiting the roundabout and staying in it. Moreover, if exiting the roundabout, the cars are predicted to exit on their right, which implies that the modalities predicted by our model are plausible and reasonable.

## 5.3 NBA dataset: sports games

We also predicted the future 10 time steps (4.0s) based on the historical 5 time steps (2.0s). The comparison of quantitative results is shown in Table 3, where the unit of reported minADE$_{20}$ and minFDE$_{20}$ is meters in the world coordinates. Note that we included both players and the basketball in the experiments. The players are divided into two different types according to their teams. The

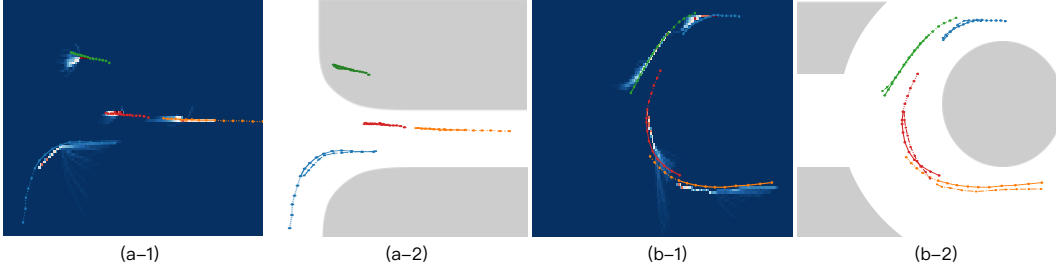

Figure 4: Qualitative results of testing cases of H3D dataset. Dashed lines are historical trajectories, solid lines are ground truth, and dash-dotted lines are prediction hypothesis. White areas represent drivable areas and gray areas represent sidewalks. We plotted the prediction hypothesis with the minimal ADE, and the heatmap to represent the distributions. (a) Intersection; (b) Roundabout.

basketball players are highly interactive and behaviors often change suddenly due to the reaction to other players. The baselines all consider the relations and interactions among agents with different strategies, such as soft attention mechanisms, social pooling layers, and graph-based representation. Owing to the dynamic interaction modeling by evolving interaction graph, our method achieves *significantly* better performance than state-of-the-art, which reduces the 4.0s minADE$_{20}$ / minFDE$_{20}$ by 22.1% / 18.1% with respect to the best baseline (Social-STGCNN). Qualitative results and analysis can be found in Section 7.3 in the supplementary materials.

### 5.4 SDD dataset: university campus

We predicted the future 12 time steps (4.8s) based on the historical 8 time steps (3.2s). The comparison of quantitative results is shown in Table 4, where the unit of reported minADE$_{20}$ and minFDE$_{20}$ is pixels in the image coordinates. Note that we included all the types of agents (e.g. pedestrians, cyclists, vehicles)in the experiments, although most of them are pedestrians. Our proposed method achieves the best performance. The 4.8s minADE$_{20}$ / minFDE$_{20}$ are reduced by 26.1% / 26.8% compared to the best baseline approach (STGAT).

## 6  Conclusions

In this paper, we present a generic trajectory forecasting framework with explicit relational reasoning among multiple heterogeneous, interactive agents with a graph representation. Multiple types of context information (e.g. static / dynamic, scene images / point cloud density maps) can be incorporated in the framework together with the trajectory information. In order to capture the underlying dynamics of the evolution of relational structures, we propose a dynamic mechanism to evolve the interaction graph, which is trained in two consecutive stages. The double-stage training mechanism can both speed up convergence and enhance prediction performance. The method is able to capture the multi-modality of future behaviors. The framework is validated by synthetic physics simulations and multiple trajectory forecasting benchmarks for different applications, which achieves state-of-the-art performance in terms of prediction accuracy. For the future work, we will handle the prediction task involving a time-varying number of agents with an extended adaptive framework. EvolveGraph can also be applied to find the underlying patterns of large-scale interacting systems which involve a large number of entities, such as very complex physics systems.

## Broader Impact

In this work, the authors introduce EvolveGraph, a generic trajectory prediction framework with dynamic relational reasoning, which can handle evolving interacting systems involving multiple heterogeneous, interactive agents. The proposed framework could be applied to a wide range of applications, from purely physical systems to complex social dynamics systems. In this paper, we demonstrate some illustrative applications to physics objects, traffic participants, and sports players. The framework could also be applied to analyze and predict the evolution of larger interacting systems, such as social networks and traffic flows. Although there are existing works using graph neural

networks to handle trajectory prediction tasks, here we emphasize the impact of using our framework to recognize and predict the evolution of the underlying relations. With accurate and reasonable relational structures, we can forecast or generate plausible system behaviors, which help much with optimal decision making. However, if the predicted relational structures are wrong or misleading, the prediction performance may be degraded since the forecast highly depends on relational structures. There is no guarantee that such frameworks are able to work well on all kinds of applications. Therefore, users are expected to assess the applicability and risk for a specific purpose.

## Footnotes

*indicates equal contribution

†Work done during Jiachen's internship at Honda Research Institute, USA.

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
