[Supplementary Material]

# 7 Additional Experimental Results and Further Analysis

In this section, we provide further experimental results and analysis, including ablative analysis, analysis on the selection of edge types and re-encoding gap, as well as additional qualitative results.

## 7.1 Ablative Analysis

We conducted ablative analysis on the benchmark datasets to demonstrate the effectiveness of heterogeneous node types, dynamically evolving interaction graph and two-stage graph learning. The best $minADE_{20}$ / $minFDE_{20}$ of each model setting are shown in the right parts of Table 2, Table 3 and Table 4. The descriptions of each model setup are provided in **Section 8.2**.

• **SG (same node type) v.s. SG**: We show the effectiveness of the distinction of agent node types. According to the prediction results in Table 2, Table 3 and Table 4, utilizing distinct agent-node embedding functions for different agent types achieves consistently smaller $minADE_{20}$ / $minFDE_{20}$ than a universal embedding function. The reason is that different types of agents have distinct behavior patterns or feasibility constraints. For example, the trajectories of on-road vehicles are restricted by roadways, traffic rules and physical constraints, while the restrictions on pedestrian behaviors are much fewer. Moreover, since vehicles usually have to yield pedestrians at intersections, it is helpful to indicate agent types explicitly in the model. With differentiation of agent types, the 4.0s $minADE_{20}$ / $minFDE_{20}$ are reduced by 4.7% / 5.8% on the H3D dataset, 8.6% / 5.8% on the NBA dataset. The 4.8s $minADE_{20}$ / $minFDE_{20}$ are reduced by 8.9% / 9.9% on the SDD dataset.

• **SG v.s. RNN re-encoding v.s. DG (double stage)**: We compare the performance of our method and SG / RNN encoding baselines. It is shown that the improvement of RNN re-encoding is limited and the prediction errors are slightly smaller than SG. Although both RNN re-encoding baseline and our method attempt to capture the potential changes of underlying interaction graph at each time step, our method achieves a significantly smaller prediction error consistently. A potential reason is despite that the RNN re-encoding process is iteratively extracting the patterns from node attributes (agent states), its capability of inferring graph evolution is limited.

• **DG (single stage) v.s. DG (double stage)**: We show the effectiveness and necessity of double-stage dynamic graph learning. It is shown that the double-stage training scheme leads to remarkable improvement in terms of $minADE_{20}$ / $minFDE_{20}$ on all three datasets. During the first training stage, the encoding / decoding functions are well trained to a local optimum, which is able to extract a proper static interaction graph. According to empirical findings, the encoding / decoding functions are sufficiently good as an initialization for the second stage training after several epochs' training. During the second training stage, the encoding / decoding functions are initialized from the first stage and finetuned, along with the training of graph evolution GRU. This leads to faster convergence and better performance, since it may help avoid some bad local optima at which the loss function may be stuck if all the components are randomly initialized. With the same hyperparameters, the single-stage / double-stage training took about 25 / 14 epochs to reach their smallest validation loss on the NBA dataset and 41 / 26 epochs on the H3D dataset. Compared to single-stage training, the 4.0s $minADE_{20}$ / $minFDE_{20}$ of double-stage training are reduced by 12.0% / 15.5% on the NBA dataset and 9.4% / 12.2% on the H3D dataset. The 4.8s $minADE_{20}$ / $minFDE_{20}$ are reduced by 13.7% / 13.9% on the SDD dataset.

## 7.2 Analysis on Edge Types and Re-encoding Gap

We also provide a comparison of $minADE_{20}$ / $minFDE_{20}$ (in meters) and testing running time on the NBA dataset to demonstrate the effect of different numbers of edge types and re-encoding gaps. In Figure 5(a), it is shown that as the number of edge type increases, the prediction error first decreases to a minimum and then increases, which implies too many edge types may lead to overfitting issues, since some edge types may capture subtle patterns from data which reduces generalization ability. The cross-validation is needed to determine the number of edge types. In Figure 5(b), it is illustrated that the prediction error increases consistently as the re-encoding gap raises, which implies more frequent re-identification of underlying interaction pattern indeed helps when it evolves along time. However, we need to trade off between the prediction error and testing running time if online prediction is required. The variance of $minADE_{20}$ / $minFDE_{20}$ in both figures are small, which implies the model performance is stable with random initialization and various settings in multiple experiments.

Figure 5: The comparison of minADE$_{20}$ / minFDE$_{20}$ (in meters) and testing running time of different model settings on the NBA dataset. We trained three models for each setting to illustrate the robustness of the method. (a) Different numbers of edge types; (b) Different re-encoding gaps. The testing running time is re-scaled to [0,1] for better illustration.

## 7.3 Additional Qualitative Results and Analysis

Figure 6 and Figure 7 show more visualizations of testing results on the H3D and NBA datasets. First, we tell that in such cases the ball follows a player at most times, which implies that the predicted results represent plausible situations. Second, most prediction hypotheses are very close to the ground truth, even if some predictions are not similar to the ground truth, they represent a plausible behavior. Third, the heatmaps show that our model can successfully predict most reasonable future trajectories and their multi-modal distributions. More specifically, in the first case of Figure 7, for the player of the green team in the middle, the historical steps move forward quickly, while our model can successfully predict that the player will suddenly stop, since he is surrounded by many opponents and he is not carrying the ball. In the second case of Figure 7, our model shows that three pairs of players from different teams competing against each other for chances. the defensing team is closer to the basket. and the player carrying the ball is running quickly towards the basket. Two opponents are trying to defend him. Such case is a very common situation in basketball games. In general, not only does our model achieve high accuracy, it can also understand and predict most moving, stopping, offending and defensing behaviors in basketball games.

## 8 Further Experimental Details

In this section, we provide important further details of experiments, which includes dataset generation, baseline approaches, as well as implementation details.

### 8.1 Datasets

#### 8.1.1 Synthetic Particle Simulations

We designed a synthetic particle simulation to validate the performance of our model. In this simulation, we have $n$ particles in an x-y plane and all the locations of particles are randomly initialized on the $y > 0$ half plane. The movement of these particles contains two phases, corresponding to two interaction graphs. Initially, particles are rigidly connected to each other and form a "star" shape. More specifically, there is a virtual centroid and each particle is rigidly connected to the centroid. it is equivalent to using a stick to connect the particle and the virtual centroid. And the distance between a certain particle and the centroid keeps the same. The particles are uniformly distributed around the centroid, which means the angle between two adjacent "sticks" is $\frac{2\pi}{n}$. In the first phase, particles move as a whole, with both translational and rotational motions. The velocity and angular velocity of the whole system are randomly initialized in a certain range. Once any one of the particles reaches $y = 0$ (the switching criterion), the movement of all the particles will transfer to the second phase. In the second phase, particles are no longer connected to each other. The motion of one particle will by

no means affect the motion of the others. In other words, each particle keeps uniform linear motions once the second phase begins. We generated 50k sample in total for training, validation and testing.

### 8.1.2 Benchmark Datasets

- H3D [31]: A large scale full-surround 3D multi-object detection and tracking dataset, which provides point cloud information and trajectory annotations for heterogeneous traffic participants (e.g. cars, trucks, cyclists and pedestrians). We selected 90k samples in total for training, validation and testing.

- NBA: A trajectory dataset collected by NBA with the SportVU tracking system, which contains the trajectory information of all the ten players and the ball in real games. We randomly selected 50k samples in total for training, validation and testing.

- SDD [33]: A trajectory dataset containing a set of top-down-view images and the corresponding trajectories of involved entities, which was collected in multiple scenarios in a university campus full of interactive pedestrians, cyclists and vehicles. We randomly selected 50k samples in total for training, validation and testing.

## 8.2 Baseline Methods

We compared the performance of our proposed approach with the following baseline methods. Please refer to the reference papers for more details.

### 8.2.1 For Synthetic Particle Simulations

- Corr. (LSTM): The baseline method for edge prediction in [18].
- LSTM (single) / LSTM (joint): The baseline methods for state sequence prediction in [18].
- NRI (static): The NRI model with static latent graph [18].
- NRI (dynamic): The NRI model with latent graph re-evaluation at each time step [18].

### 8.2.2 For Benchmark Datasets

- Social-LSTM [1]: The model encodes the trajectories with an LSTM layer whose hidden states serve as the input of a social pooling layer.

- Social-GAN [7]: The model introduces generative adversarial learning scheme into S-LSTM to improve performance.

- Social-Attention [42]: The model deals with spatio-temporal graphs with recurrent neural networks, which is based on the architecture of Structural-RNN [15].

- Gated-RN [5]: The model infers relational behavior between road users and the surrounding environment by extracting spatio-temporal features.

- Trajectron++ [35]: The approach represents a scene as a directed spatio-temporal graph and extract features related to the interaction. The whole framework is based on conditional variational auto-encoder.

- NRI [18]: The model is formulated as a variational inference task with an encoder-decoder structure. This is the most related work.

- STGAT [14]: The model is a variant of graph attention network, which is applied to spatio-temporal graphs.

- Social-STGCNN [29]: The model is a variant of graph convolutional neural network, which is applied to spatio-temporal graphs.

### 8.2.3 Ablative Baselines

- SG (same node type): This is the simplest model setting, where only a static interaction graph is extracted based on the history information. The same node embedding function is shared among all the agent nodes.

- SG: This setting is similar to the previous one, except that different node embedding functions are applied to different types of agent nodes.

Figure 6: Additional qualitative results of the H3D dataset. The upper figures are the visualization of predicted distributions, and the lower figures are the best prediction hypotheses. The white areas are drivable areas and gray areas are sidewalks. Note that there are agents moving on the sidewalks since we also include pedestrians and cyclists.

- RNN re-encoding: The interaction graph is re-encoded every $\tau$ time steps using an RNN encoding process. Note that this is different from our model, since the RNN encoding process only captures evolution of node attributes without explicitly modeling the dependency of consecutive underlying interaction graphs.
- DG (single stage): This is our whole model, where the encoding, decoding functions and the graph evolving GRU are all trained together from scratch.
- DG (double stage): This is our whole model with double stage interaction graph learning, where the encoding, decoding functions trained at the first stage are employed as an initialization in the second stage.

## 8.3 Implementation Details

For all the experiments, a batch size of 32 was used and the models were trained for up to 20 epochs during the static graph learning stage and up to 100 epochs during the dynamic graph learning stage

Figure 7: Additional qualitative results of the NBA dataset. The upper figures are the visualization of predicted distributions, and the lower figures are the best prediction hypotheses. The line colors indicate teams and blue lines are the trajectories of basketball.

with early stopping. We used Adam optimizer with an initial learning rate of 0.001. The models were trained on a single TITAN Xp GPU. We used a split of 65%, 10%, 25% as training, validation and testing data.

Specific details of model components are introduced below:

- *Agent node embedding function*: for each different node type, a distinct two-layer gated recurrent unit (GRU) with hidden size = 128.
- *Context node embedding function*: four-layer convolutional blocks with kernel size = 5 and padding = 3. The structure is [[Conv, ReLU, Conv, ReLU, Pool], [Conv, ReLU, Conv, ReLU, Pool]].
- *Agent node update function*: a three-layer MLP with hidden size = 128.
- *Edge update function*: for both agent-agent edges and agent-context edges, a distinct three-layer MLP with hidden size = 128.
- *Encoding function*: a three-layer MLP with hidden size = 128.
- *Decoding function*: a two-layer gated recurrent unit (GRU) with hidden size = 128.

- *Recurrent graph evolution module*: a two-layer GRU with hidden size = 256.

Specific experimental details of different datasets are introduced below:

- *Synthetic simulations*: 2 edge types, re-encoding gap = 1, encoding horizon = 20.
- *H3D dataset*: 5 edge types, re-encoding gap = 5, encoding horizon = 5.
- *NBA dataset*: 6 edge types, re-encoding gap = 4, encoding horizon = 5.
- *SDD dataset*: 4 edge types, re-encoding gap = 5, encoding horizon = 5.

## 8.4 Illustrative Diagram of the Decoding Process

We provide an illustrative diagram of the decoding process, which is shown in Figure 8. In this figure, without loss of generality we demonstrate the decoding process for only one node in a five-node observation graph to illustrate how the decoding process works. Figure 8(a) shows the observation graph, we choose the node on the right as an example. Figure 8(b) shows the process of using MLPs to process a specific edge, where $z_{ij,l}, l = 1, 2, ..., L$ denotes the probability of the edge belonging to a certain edge type $l$. The processed edges are shown in red. Figure 8(c) shows the sum over every incoming edge attribute of this node. Then we input the result into the decoding GRU. The decoding GRU outputs several Gaussian components and their corresponding weights. We sample one specific Gaussian component based on the weights. Then we use the $\boldsymbol{\mu}$ of the sampled Gaussian distribution as the output state at this step. $\boldsymbol{\mu}$ is used as the input into the next decoding step (if it's not the burn-in step). We iterate the decoding process several times until the desired prediction horizon is reached.

Figure 8: An illustrative diagram of the decoding process.