[Reviews · NeurIPS 2020]

Review 1

Summary and Contributions: The authors propose a novel relational reasoning model that, unlike previous work, infers both the graph structure and updated node parameters at each time step (after a burn-in period). I think there are two contributions in this paper: 1. the graph structure of the system at hand should predicted dynamically rather than assumed fixed and given or fixed but learned 2. A specific method to estimate

Strengths: Predicting both updated node attributes and graph structure is a neat idea. Relational reasoning has received considerable attention in recent years at NeurIPS and I think this contribution is novel and significant. I appreciated how the authors managed to separate their contributions throughout the paper. In particular, as some of the baselines can be adapted to work dynamically (e.g. NRI can just refresh the inferred graph structure at every step after a burn-in period), separating the general idea of dynamically predicting the graph structure and the specific way of doing so was very useful.

Weaknesses: Inferring the graph structure might be especially beneficial for relatively large systems, where methods based on over-connected graphs might struggle to scale. I would love to see an experiment on a system with many nodes (see e.g. Learning to Simulate Complex Physics with Graph Networks). It would be really good to add some comments on what structure is actually predicted. Especially on large systems (e.g. Learning to Simulate Complex Physics with Graph Networks) one could start to make some serious analysis of the predictions themselves and how they are influenced by the context (e.g. an particle cloud representing an object becomes connected to another object upon contact).

Correctness: Yes

Clarity: Yes, some of the notation reinvents the wheel a bit and could rest on the previous work from e.g. Relational inductive biases, deep learning, and graph networks, but it's a minor thing.

Relation to Prior Work: Yes

Reproducibility: Yes

Additional Feedback: Very minor suggestion: I am not sure "computer vision" is the correct sub-field for this work. I think this will affect which poster session you are sent to, and therefore who comes to your poster. Take a quick look if there is a more appropriate sub-field if you have time. Maybe: "Algorithms -> Relational Learning; Deep Learning -> Recurrent Networks"? Thank you for sharing these cool ideas, I hope my suggestions help. ============ After Author Feedback =========== Thank you for taking the time to write a rebuttal, and for sharing these cool ideas. I stand by my initial assessment that this paper should be accepted. Best.


Review 2

Summary and Contributions: This paper proposes a framework for trajectory prediction for multiple interactive agents based on latent interaction graphs. This framework can be used exploit multiple modal data capture the multi-modality of future behaviours. They propose a double-stage training pipeline that improves training efficiency, and model performance.

Strengths: 1. From my understanding, their main contributions are two part: 1) their framework can exploit multimodal data for behaviours modelling 2) they propose a dynamic mechanism and a double stage training pipeline to improve performance. 2. The experiments are extensive. Their EvolveGraph outperforms other baselines.

Weaknesses: 1. Some parts of this paper are hard to understand, for example, Section 3 and 4. 2. There is no discussion on why their proposed dynamic mechanism and double stage training pipeline improve the performance. I would suggest to include some intuitive explanation and empirical evidence in introduction or it is hard to convince readers to accept the superiority of your model. For me, the novelty of introducing these two modules and integrating multimodal data is still limited for a top machine learning conference.

Correctness: The claims and method seem correct.

Clarity: Most parts are well-written but some parts are hard to understand.

Relation to Prior Work: They have discussed the previous work in related work and intro.

Reproducibility: Yes

Additional Feedback: I read the authors' response and appreciate the authors' efforts on improving this paper. Some of my concerns have been addressed. Integrating multi-modalities into graph learning could be an interesting contribution to this community. Thus, I upgraded my score to 6.


Review 3

Summary and Contributions: The authors propose a method for multi-agent trajectory prediction. In the prediction framework, it models interaction of multi-agents with a structure called EvolveGraph which could be dynamically updated. With a double-stage training pipeline, EvolveGraph firstly extracts interaction patterns, called static interaction graph, from the observed trajectories and context information. And then the static interaction graph could be evolved to reflect the dynamic relations of agents.

Strengths: Using graph to represent the interaction of multi-agents is one of the new trends in trajectory prediction area.

Weaknesses: The paper presents several critical issues. 1. Although double stage results are better than single stage results, their ADE/FDE are very close in table2, 3 and 4. Since double stage has dynamic interaction graph, it may has a strong advantage in relatively long time prediction. Dynamic interaction graph is the main innovation point of this work, it would be better EvolveGraph could be evaluated and compared in a long time prediction task. 2. It looks nothing new in the aspect of multi-modality.

Correctness: It seems correct.

Clarity: Well written and easy to follow.

Relation to Prior Work: No. Lack of discussion and compare with other graph based trajectory prediction methods.

Reproducibility: Yes

Additional Feedback:


Review 4

Summary and Contributions: This paper is an extension of the Neural Relational Inference framework for relational reasoning and trajectory prediction. It extends NRI in two key ways: first, relations are re-predicted periodically, leading to a dynamic relation graph. Second, the model ihs trained to predict a mixture of distributions, which allows for multimodal trajectory prediction. The approach is compared against NRI as well as other trajectory prediction methods to demonstrate its approved ability to recover known relations over NRI as well as its ability to predict future trajectories better than all compared methods.

Strengths: The work is very well-motivated and explained in a clear manner. It is sensible to extend NRI to dynamic relation graphs as well as to produce multi-modal predictions, and the empirical evaluation demonstrates that the approach is practically useful in addition to being conceptually interesting. Furthermore, the experiments are extensive - EvolveGraph is compared against not only all of the relevant NRI baselines but also a number of trajectory prediction models that include relational reasoning. A number of ablations are given in the supplementary material which demonstrate the impact of various modeling choices, which is very useful information for a practitioner.

Weaknesses: The ability of EvolveGraph to uncover known dynamic relations is not explored in as much detail as it could be. More specifically, the one synthetic experiment designed to evaluate this is somewhat simple, in that all relations change from "active" to "inactive" for all entities at the same moment in time, and this switch happens once. What happens when relations change at different times for different variables? What happens if the re-encoding gap is "out of sync" with the actual change in relations? How well does the model perform if relations change multiple times aperiodically? These questions are not explored here. There are a few modeling decisions which are made that are not explained or explored either. The ones that stick out to me: - The observation model has learned attentional coefficients that seem to be static across time. Do these contribute meaningfully to model performance? Also, doesn't the fact that these coefficients are static mean that they "pre-determine" the impact some variables have on others in a data-agnostic manner? - A different prediction mode is selected for each variable for every time step. What happens if modes are re-evaluated less often? How do the frequency of mode selection and relation re-prediction relative to each other impact final performance? - How many modes does the model predict, and how does performance vary as the number of predicted modes changes? Right now, it's difficult to understand if the performance improvements are primarily due to modeling multi-modality, modeling dynamic relations, or both. These criticisms are relatively minor, however; there is enough present in this work for it to be a worthwhile publication.

Correctness: Yes, the paper and methodology are correct.

Clarity: The paper is very easy to understand and explained in sufficient detail for the most part. The attention coefficients I mentioned previously are not explained in sufficient detail. Also, the loss equation used for training is not in the main paper but instead deferred to the supplementary, when it would be much clearer to include it in the appropriate section in the main paper.

Relation to Prior Work: The connection between this work and NRI is made very clearly, and other appropriate baselines are also used for experimentation.

Reproducibility: Yes

Additional Feedback: -Equations 8-10 are somewhat redundant with eqs. 5-7. These can be compressed to save space/make it clear that the same overall process is happening during both phases but using different inputs. This space would be better used to move the loss function equations into the main paper. -Unlike NRI, EvolveGraph was not trained as a VAE - specifically, no prior was enforced on the relations. What impact does using a prior here have on model performance? -------------------------------------------------------------------------------------------------------------- POST AUTHOR RESPONSE EDIT: Thanks for the response. I still think this paper is very interesting and should definitely be accepted.

[Author Response · NeurIPS 2020]

Table 1: Quantitative Results Compared with Additional Graph Based Methods (**R3**)

| Dataset | H3D | | | | NBA | | | | SDD |
|---|---|---|---|---|---|---|---|---|---|
| Time | 1.0s | 2.0s | 3.0s | 4.0s | 1.0s | 2.0s | 3.0s | 4.0s | 4.8s |
| STGAT (ICCV 2019) | 0.24 / 0.33 | 0.34 / 0.48 | 0.46 / 0.77 | 0.60 / 1.18 | 0.62 / 1.04 | 1.26 / 1.91 | 1.73 / 3.05 | 2.47 / 3.86 | 18.8 / 31.3 |
| STGCNN (CVPR 2020) | **0.23** / 0.32 | 0.36 / 0.52 | 0.49 / 0.89 | 0.73 / 1.49 | 0.68 / 1.12 | 1.34 / 2.13 | 1.85 / 3.10 | 2.56 / 4.03 | 20.6 / 33.1 |
| EvolveGraph | **0.23 / 0.29** | **0.31 / 0.44** | **0.39 / 0.58** | **0.48 / 0.86** | **0.56 / 0.80** | **0.82 / 1.19** | **1.20 / 1.92** | **1.76 / 3.04** | **13.9 / 22.9** |

The authors thank the reviewers for their helpful and valuable feedback. We are encouraged that the reviewers
pointed out the strengths of our paper: novelty and significance (R1,R4); strong motivation (R1,R4); correctness and
reproducibility (R1,R2,R3,R4); clarity (R1,R3,R4); extensive experiments and analysis (R1,R2,R4); broader impact
(R1-R4). We will respond to the feedback from all reviewers below and incorporate all feedback in the revised version.
**R1**: The authors sincerely express our acknowledgement for your helpful suggestions and affirmation on the novelty
and significance of our work. We agree that inferring the graph structure is definitely crucial especially for very large
interacting systems with many nodes, and we will add the suggested reference to the main paper. Some comments on
the predicted structures on physics system can be found in Fig. 2 and L247-250. Due to the limit of computational
resources and time, we were not able to provide comprehensive results on very large systems. However, it will definitely
be a meaningful future research direction to apply the proposed framework to large interacting systems such as very
complex physics systems. Moreover, we will figure out the most appropriate sub-fields for this paper as you mentioned.
**R2**: The authors are happy to refine Sec. 3 and 4 if R2 could provide detailed advice, since all the other reviewers think
the paper is easy to follow. Some intuitive explanation and empirical evidence can be found in L32-42 and supp L26-39.
**R2,R3,R4**: **Multi-modality and dynamic relations** The multi-modality is discussed in detail from three aspects in
L188-203. In particular, we would like to highlight the multi-modality driven by dynamic relation modeling. Different
evolution of relations recurrently leads to more diverse outcomes, which in return establishes the value of dynamic
relations in multi-modality. Therefore, we would say that the performance improvements come from both dynamic
relations and multi-modality, which are highly correlated. We believe dynamically evolving the interaction graph
structures and inherently encouraging diversity and multi-modality, to our best knowledge, is novel and significant for
trajectory prediction as mentioned by R1 and R4.
**R3**: **Comparison with graph-based methods** The authors thank the reviewer for pointing out additional related
work. In fact, we have compared EvolveGraph with two graph-based methods in Table 2-4, i.e. NRI (ICML2018)
and Trajectron++ (ECCV2020). EvolveGraph can reduce 4.0s ADE/FDE by 40.5%/42.2% on NBA (L292-295), and
reduce 4.8s ADE/FDE by 34.4%/36.6% on SDD (L300-301). We also add requested comparison results of another two
graph-based methods STGAT (ICCV2019) and Social-STGCNN (CVPR2020) in **Table 1 in the rebuttal**. Compared
to STGAT, EvolveGraph can reduce 4.0s ADE/FDE by 20.0%/27.1% on H3D, 28.7%/21.2% on NBA, and reduce 4.8s
ADE/FDE by 26.1%/26.8% on SDD. We will add these results and discussion in the revised version.
**Dynamic interaction graph** The authors would like to clarify that both DG (single stage) and DG (double stage) have
dynamic interaction graphs, while SG only has static interaction graph (supp L128-142). The difference between DG
(single stage) and DG (double stage) mainly lies in the training procedures. Detailed analysis of ablative results was
provided in supp L4-39 and we highlight some facts and statements here. In Table 2-4, compared to SG, DG (double
stage) can reduce 4.0s ADE/FDE by 21.3%/24.6% on H3D, 31.3%/33.3% on NBA, and reduce 4.8s ADE/FDE by
32.5%/37.1% on SDD. The prediction horizons are standard in trajectory prediction tasks and we believe empirical
results on all datasets already validate the significant superiority of dynamic graph.
**R4**: **Uncover dynamic relations** We appreciate the suggestion of additional synthetic experiments. (a) With "out of
sync", we observed that smaller re-encoding gap is beneficial to evaluate the relation changes. Although there is a
trade-off between the performance and computational cost, prediction error decreased as re-encoding gap becomes
smaller. (b) Due to the time limit, we were not able to conduct other experiments. However, we believe that our three
real-world prediction tasks can be used to evaluate the model performance with multiple aperiodic relation changes
to some extent. For example, in NBA dataset the relations among players change frequently and aperiodically due to
intensive cooperations and competitions. Our significant improvements in accuracy demonstrate the efficacy of dynamic
relation modeling. **Attention coefficients** The authors would like to clarify that the attention coefficient $\alpha_{ij}$ is not a
simple learnable constant, but it is computed based on the features of involved agents $i$ and $j$ similar to GAT [A]. As
the encoding horizon slides to future steps, the coefficients are re-computed at each time. Therefore, these coefficients
are not data-agnostic and they can figure out relative significance of impacts during each re-encoding process. We will
make it clear in the revised version. [A] Veličković, et al. "Graph Attention Networks." ICLR, 2018.
**Modes and dynamic relations** The authors would like to clarify that we do not either pre-define a fixed number of
modes, or explicitly differentiate multiple modes. Instead, at each time our model outputs Gaussian mixture distributions
and the predicted trajectories are sampled from these distributions. Together with interaction graph evolution, this
process may naturally result in multiple modes without having an explicit mode selection process. Please also refer to
L14-19 above for more explanations. **Loss function** The camera-ready version allows an additional content page and
we will move the loss function to the main paper. **Prior** Since the underlying relations in real-world heterogeneous
interacting systems are complex and diverse, it is not easy to select a proper prior of dynamic relations for specific tasks.
Some prior with preference on "non-edge" may encourage sparsity for easier visualization or interpretability. Exploring
prior incorporation and its detailed impact will be our future work.

[Meta-Review · NeurIPS 2020]

Reviewers agree that the work is interesting and novel, and many of the concerns raised in the reviews were addressed by the authors in their rebuttal. The multi-modal aspects are applied sensibly, although perhaps slightly oversold.